# A Prospective Multicenter Study of “Inside Stents” for Biliary Stricture: Multicenter Evolving Inside Stent Registry (MEISteR)

**DOI:** 10.3390/jcm10132936

**Published:** 2021-06-30

**Authors:** Hirofumi Kogure, Hironari Kato, Kazumichi Kawakubo, Hirotoshi Ishiwatari, Akio Katanuma, Yoshinobu Okabe, Toru Ueki, Tesshin Ban, Keiji Hanada, Kazuya Sugimori, Yousuke Nakai, Hiroyuki Isayama

**Affiliations:** 1Department of Gastroenterology, Graduate School of Medicine, The University of Tokyo, Tokyo 113-8655, Japan; kogureh.tky@gmail.com (H.K.); ynakai.tky@gmail.com (Y.N.); 2Department of Gastroenterology and Hepatology, Okayama University Graduate School of Medicine, Dentistry and Pharmaceutical Sciences, Okayama 700-8558, Japan; drkatocha@yahoo.co.jp; 3Department of Gastroenterology and Hepatology, Hokkaido University Graduate School of Medicine, Sapporo 060-8648, Japan; nroopy@gmail.com; 4Department of Medical Oncology and Hematology, Sapporo Medical University School of Medicine, Sapporo 060-8543, Japan; ishihiro481019@gmail.com; 5Center for Gastroenterology, Teine-Keijinkai Hospital, Sapporo 006-0811, Japan; akatanuma@gmail.com; 6Division of Gastroenterology, Department of Medicine, Kurume University of Medicine, Kurume 830-0011, Japan; yib01074@nifty.com; 7Department of Internal Medicine, Fukuyama City Hospital, Fukuyama 721-8511, Japan; torueki0730@yahoo.co.jp; 8Department of Gastroenterology, Nagoya Daini Red Cross Hospital, Nagoya 466-8650, Japan; vanzosan@gmail.com; 9Department of Gastroenterology, Onomichi General Hospital, Onomichi 722-8508, Japan; kh-ajpbd@nifty.com; 10Division of Gastroenterology, Yokohama City University Graduate School of Medicine, Yokohama 232-0024, Japan; sugimori@yokohama-cu.ac.jp; 11Department of Endoscopy and Endoscopic Surgery, Graduate School of Medicine, The University of Tokyo, Tokyo 113-8655, Japan; 12Department of Gastroenterology, Graduate School of Medicine, Juntendo University, Tokyo 113-8431, Japan

**Keywords:** biliary stricture, endoscopic biliary stent placement, inside stent

## Abstract

Background: Endoscopic biliary stent placement is the standard of care for biliary strictures, but stents across the papilla are prone to duodenobiliary reflux, which can cause stent occlusion. Preliminary studies of “inside stents” placed above the papilla showed encouraging outcomes, but prospective data with a large cohort were not reported. Methods: This was a prospective multicenter registry of commercially available inside stents for benign and malignant biliary strictures. Primary endpoint was recurrent biliary obstruction (RBO). Secondary endpoints were technical success of stent placement and removal, adverse events, and stricture resolution. Results: A total of 209 inside stents were placed in 132 (51 benign and 81 malignant) cases with biliary strictures in 10 Japanese centers. During the follow-up period of 8.4 months, RBO was observed in 19% of benign strictures. The RBO rate was 49% in malignant strictures, with the median time to RBO of 4.7 months. Technical success rates of stent placement and removal were both 100%. The adverse event rate was 8%. Conclusion: This prospective multicenter study demonstrated that inside stents above the papilla were feasible in malignant and benign biliary strictures, but a randomized controlled trial is warranted to confirm its superiority to conventional stents across the papilla.

## 1. Introduction

Endoscopic biliary stent placement is the standard of care for both malignant and benign biliary strictures. Since the first application of biliary stents [1], duodenobiliary reflux is considered the cause of stent clogging due to bacterial biofilm [2]. Various efforts have been made to prevent stent occlusion, i.e., medications such as antibiotics [3] and ursodeoxycholic acid [4], self-expandable metal stents with a large diameter [5], drug-eluting stents [6], and anti-reflux stents [7]. There are two treatment options to prevent duodenobiliary reflux itself; anti-reflux stents and stent above the papilla. Clinical trials of anti-reflux biliary stents have been conducted, but their role is still controversial [8]. Stents above the papilla, so-called “inside stents,” have been investigated using both metal and plastic stents [9]. Fully-covered self-expandable metal stents (FCSEMS) with a retrieval lasso are increasingly utilized for benign biliary strictures at the hilum or anastomotic strictures [10,11,12,13], but FCSEMS is not always indicated in patients with non-dilated ducts. Since introducing the concept of inside stents using plastic stents, there are some concerns about stent removal and exchange, too [14]. Encouraging data of inside stents were reported, using hand-made plastic stents with a retrieval thread [15,16,17,18,19,20]. Recently, inside stents with a retrieval thread are commercially available in Japan [21], and we conducted this prospective multicenter cohort study to evaluate the safety and effectiveness of these commercially available inside stents in patients with both malignant and benign biliary strictures.

## 2. Methods

### 2.1. Study Design

This was a prospective, multicenter registry study conducted in 10 Japanese centers. Consecutive patients aged 20 years or older who underwent “inside stent” placement via ERCP to manage benign or malignant biliary stricture were enrolled in the study. Exclusion criteria were biliary strictures within 2 cm from the ampulla, tumor invasion to the ampulla, a previous history of gastrectomy with Roux-en-Y or Billroth II reconstruction, or concomitant acute pancreatitis at the time of ERCP were excluded. The study was approved by the ethical committee at each center, and written informed consent was obtained from all patients. The study was registered UMIN clinical trial registration (UMIN000013291).

### 2.2. Placement and Removal of Inside Stents

Plastic stents used in this study are Amsterdam-type polyethylene stents, preloaded with their stent delivery system and can be repositioned until fully deployed (ThroughPass IS, Gadelius Medical K.K., Tokyo, Japan, Figure 1). The proximal end has a flap to prevent distal migration. While the distal end has no flaps, a knotted nylon thread was attached to the distal end to allow stent retrieval. The stent size used in the study is either 7-Fr or 8.5-Fr, and the length is 9-cm or 12-cm. Two types of central bends, either light or deep angle, are available and are selected according to the bile duct configuration. Inside stents were placed above the papilla with the retrieval thread in the duodenum. Multiple stents are inserted if clinically indicated. Ampullary intervention such as endoscopic sphincterotomy (ES) and endoscopic papillary balloon dilation (EPBD) was performed at the discretion of endoscopists. Additional ampullary interventions were unnecessary only for stent removal in cases without prior ampullary interventions. In cases with cholangitis at the time of RBO, the inside stent was replaced by endoscopic nasobiliary drainage (ENBD) tube first if necessary, and after resolution of cholangitis, ENBD was replaced by a new inside stent or the other type of stents at the discretion of each physician.

For stent removal, the retrieval thread was grasped with forceps, and the stent was pulled out of the bile duct. The stent was removed all the way through the scope with forceps or removed with a snare once the distal end was in the duodenum. The treatment strategy was decided at the discretion of each physician. Planned stent exchange was allowed if physicians considered it clinically appropriate.

### 2.3. Clinical Outcomes

The primary endpoint was recurrent biliary obstruction (RBO). Secondary endpoints were the technical success of stent placement and removal, adverse events, and stricture resolution. The effects of an ampullary intervention on RBO were also evaluated. Some patients received multiple sessions of inside stent placement, and all clinical outcomes were evaluated as per session analyses.

### 2.4. Definitions

Technical success of stent placement was defined as placing one or more inside stents in an appropriate position. Technical success of stent removal was defined as the removal of inside stents completely from the bile duct. RBO was defined as a composite of stent occlusion and migration according to TOKYO criteria 2014 [22]. Adverse events were graded according to the literature [23].

### 2.5. Statistical Analysis

Given the miscellaneous etiology of this cohort study, formal sample size calculation was not performed, and the target number of 200 inside stent placement was determined based on the estimated case number at each center. The numbers are shown in number (%) or median with interquartile ranges (IQRs). To compare categorical and continuous variables between groups, the Fisher’s exact test and Student’s t-test were used, respectively. Median cumulative survival time and time to RBO (TRBO) with 95% confidential interval (CI) were calculated using a Kaplan-Meier analysis and compared using the log-rank test. All statistical analyses were performed with JMP Pro version 15.1 (SAS Institute, Cary, NC, USA).

## 3. Results

### 3.1. Patients

From March 2014 to October 2016, a total of 209 inside stents were placed in 132 cases with biliary strictures. Patient characteristics are shown in Table 1. Biliary strictures were benign in 51 cases (39%) and malignant in 81 cases (61%). The most common etiology of biliary strictures was post-liver transplantation stricture (61%) among benign strictures and bile duct cancer (44%) in malignant strictures. Biliary strictures at the hilum were common both in benign (53%) and malignant (79%) biliary strictures. Prior biliary drainage was present in 35 patients (59%) with benign strictures and 34 patients (42%) with malignant strictures. The prior biliary drainage was inside stents in 31, ENBD tube in 30 patients, conventional plastic stents across the papilla in 7 patients, and percutaneous transhepatic biliary drainage in 1 patient. While planned stent exchange was performed in 80% of benign strictures, on-demand exchange at the time of stent occlusion was selected in 54% of malignant biliary strictures. No ampullary interventions such as ES and EPBD were performed in about half of patients before inside stent placement both in benign and malignant biliary strictures. The median follow-up period was 8.4 (IQR, 3.1–14.3) months. The 6-month and 1-year survival rate was 77% and 59% in malignant biliary strictures and 97% and 88% in benign biliary strictures.

### 3.2. Stent Placement 

Details of stent placement are shown in Table 2. Balloon dilation of biliary strictures to facilitate stent insertion was performed in 38% of benign strictures and 8% of malignant strictures. The stent size was 7-Fr in 47% and 45% of benign and malignant strictures, respectively. Multiple stents were placed in 60% of benign biliary strictures and 40% of malignant biliary strictures. Stent placement was successful in 100%, though technical difficulty was encountered in 3 cases (1%). One stent migrated distally during stent deployment and needed repositioning. There was one proximal stent migration of the first inside stent during the second stent placement. There was one case with the thread migrated into the bile duct during stent deployment.

### 3.3. RBO, Adverse Events, and Stricture Resolution

RBO and adverse events are summarized in Table 2. RBO developed in 20 sessions (19%) in 14 patients with benign strictures and 52 sessions (49%) in 37 patients with malignant strictures. Cumulative TRBO is shown in Figure 2. Median TRBO was 4.7 (95%CI, 2.3–6.8) months. The RBO rates at 6-month and 1-year were 54% and 79%, respectively, in malignant strictures, and 22% and 29%, respectively, in benign strictures. TRBO did not differ by ampullary interventions, both in benign and malignant strictures (Figure 3). The 6-months RBO rates in benign biliary strictures were 29%, 24%, and 23% in patients with no ampullary interventions, EPBD, and ES (*p* = 0.63). Median TRBO in malignant biliary strictures was 4.7 (95%CI, 1.8–6.8) and 4.4 (95%CI, 2.1–8.1) months in patients with ES and without ampullar interventions (*p* = 0.82).

Adverse event rates were 8% both in benign and malignant strictures. Four cases and one case of cholangitis without RBO, one case and three cases of cholecystitis, and three cases and three cases of pancreatitis occurred in both benign and malignant strictures, respectively.

Among 51 benign biliary strictures, stricture was resolved in 8 cases with the stricture resolution rate of 16%. The causes of biliary strictures were post-living donor liver transplantation in 5, primary sclerosing cholangitis in 2, and unknown in 1. The stent size was 8.5-Fr in 7 and 7-Fr in 1, and the median stent number placed was 2. Stricture resolution was obtained by a single session of inside stent placement in 2 patients, and the remaining six patients needed multiple sessions of inside stent placement for stricture resolution. The median duration from the initial stent placement was 6.4 (IQR, 4.8–7.4) months. 

Among 81 malignant biliary strictures, inside stents were placed as preoperative biliary drainage in 8 cases with bile duct cancer. Biliary strictures were located at CBD in 3 and at hilum in 5. The median time to surgery was 13 (IQR, 6–25) days. There were 2 RBOs (25%) after 3 and 7 days after stent placement and one moderate cholecystitis.

### 3.4. Stent Removal

Stent removal was attempted in 133 sessions and was successful in 100%. Reasons for stent removal were RBO in 68, planned stent exchange in 50, stricture resolution in 9, and others in 6.

The technical difficulty in stent removal was encountered in 8 sessions of 4 cases with benign biliary strictures; 3 proximal stent flaps trapped at the stricture, three thread migration into the bile duct, and two stent fractures. Inside stents were removed with ease using forceps and/or a snare in 126 cases. Intraductal insertion of forceps (*n* = 5) or basket catheter (*n* =3) was necessary for stent retrieval in cases with difficult stent removal, but no post-ERCP pancreatitis developed even after intraductal device insertion for stent removal. 

## 4. Discussion

In this prospective multicenter registry study, inside stents for malignant and benign biliary strictures were feasible both in benign and malignant strictures with technical success rates of 100% of 209 sessions. Stent removal was also successful in 100% in 134 attempts of removal, mostly by grasping the thread, allowing for stent exchange as a planned procedure or at the time of RBO. The stricture resolution rate was 16% among benign biliary strictures.

Although stent placement across the papilla has been the standard of care, the concept of metal and plastic stents above the papilla has been investigated both in malignant and benign biliary strictures [9]. However, most studies were retrospective with a small sample size, and no RCTs have proven the superiority of stents above the stricture to stents across the papilla thus far. In our study cohort, the median TRBO of inside stents was 4.7 months in malignant biliary strictures, which appear similar or longer than conventional plastic stents [24]. Bilateral metal stenting is recommended nowadays for better stent patency as well as survival in hilar malignant biliary obstruction [25]. Although recent development of dedicated metal stents allows easy stent deployment either in a stent-in-stent or side-by-side fashion [26,27,28], reinterventions can be technically demanding even by experts. Current clinical guidelines by the Japanese Society of Hepato-Biliary-Pancreatic Surgery recommend both uncovered metal stents and stent exchange using plastic stents for unresectable hilar malignant biliary stricture [29]. If stent patency is comparable to uncovered metal stents, then the easy exchange of inside stents can be beneficial since the conversion rates to percutaneous transhepatic biliary drainage (PTBD) were reportedly higher in bilateral metal stent placement despite its initial longer stent patency [30].

In addition to potentially longer TRBO of inside stents, reduced bacterial contamination might lead to a decrease in post-surgical infection when used as preoperative biliary drainage. Endoscopic nasobiliary drainage (ENBD) is recommended as preoperative biliary drainage for hilar cholangiocarcinoma because of its lesser risk of cholangitis than plastic stents, and lesser dissemination than PTBD [31]. However, prolonged ENBD tube placement would impair the quality of life, and plastic stents are often selected in clinical practice. In our study cohort, eight cases received inside stent placement as preoperative biliary drainage and two early RBOs were encountered. Due to the small number of preoperative cases in our cohort, the results were difficult to interpret, but recent pilot retrospective studies suggested inside stents can be an alternative to ENBD in this setting [32,33]. The role of inside stents as preoperative biliary drainage for hilar cholangiocarcinoma should be further evaluated in a large cohort, and post-surgical infectious complications should be included in the study outcomes.

Preservation of the sphincter of Oddi in addition to stent placement above the papilla may further avoid duodenobiliary reflux, but it is still controversial whether ES should be avoided before inside stent placement or not. In our study cohort, ampullary interventions such as ES and EPBD were performed in about half of the cases but did not affect either TRBO or post-ERCP pancreatitis. The study results are in line with a previous animal study [34], in which Sung et al. compared four groups of no intervention, sphincterotomy alone, stent above the papilla, and stent across the papilla, and after 2–6 weeks of interventions, the same bacteria were found in the gastrointestinal tract and the bile duct only in the group of the stent across the papilla. The stent placement above the papilla can also avoid stent occlusion by food debris around the distal stent end in the duodenum, which we often encounter at endoscopic reinterventions for RBO. Dietary fibers and bacterial biofilm were the major contributing factors of stent clogging [35], and reflux of large food debris into the bile duct is less likely even after ES. Since ES allows easy procedure including biliary access and stent manipulation, it should be considered when repeated ERCP procedures such as stent exchange are expected in cases with difficult biliary access.

There are some limitations to this study. The lack of a control group and inclusion of both benign and malignant biliary stricture is the major limitation. Although the theoretical advantage of inside stents is the prevention of duodenobiliary reflux, which would reduce bacterial contamination in the biliary tree and prevent cholangitis or RBO due to sludge, whether inside stents would prolong TRBO needs a prospective comparative study with a matched cohort. In addition, treatment strategy was various, such as planned stent exchange and on-demand exchange, according to the indications of biliary drainage. Furthermore, the follow-up period of 8.4 months was relatively short to evaluate the long-term outcomes, especially in cases with benign biliary strictures. Nevertheless, our study first evaluated the feasibility of commercially available inside stents with a retrieval thread in a large, multicenter cohort.

In conclusion, inside stents above the papilla were feasible in malignant and benign biliary strictures. However, due to the limitation of our single-arm study design, the concept of stents above the papilla cannot be proven in this study. A prospective randomized controlled trial is ongoing to confirm the superiority of inside stents above the papilla to conventional stents across the papilla (UMIN000036315). We also need to elucidate subgroups who would benefit from inside stents.

## Figures and Tables

**Figure 1 jcm-10-02936-f001:**
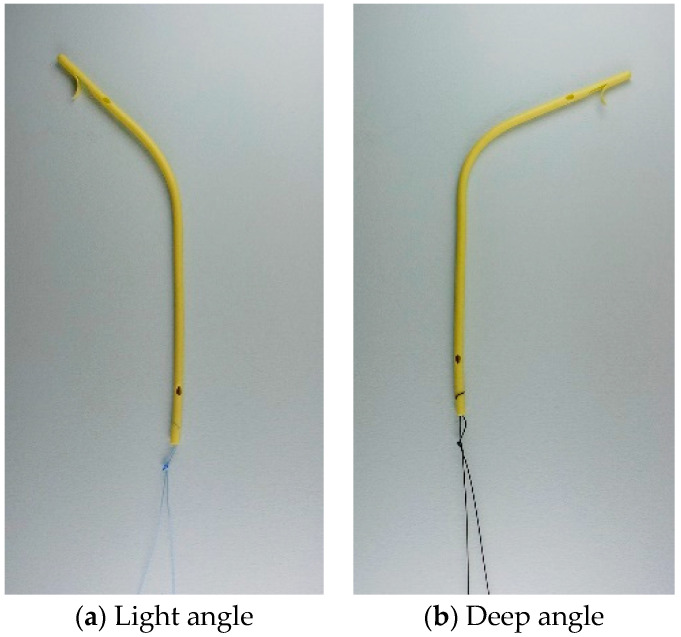
Commercially available inside stents (ThroughPass IS, Gadelius Medical K.K., Tokyo, Japan). There are two types of central bends, light angle (**a**) and deep angle (**b**). The angles of central bends were 34 degrees in the light angle and 75 degrees in the deep angle. The retrieval thread attached to the distal end was made of Nylon and 10 cm long.

**Figure 2 jcm-10-02936-f002:**
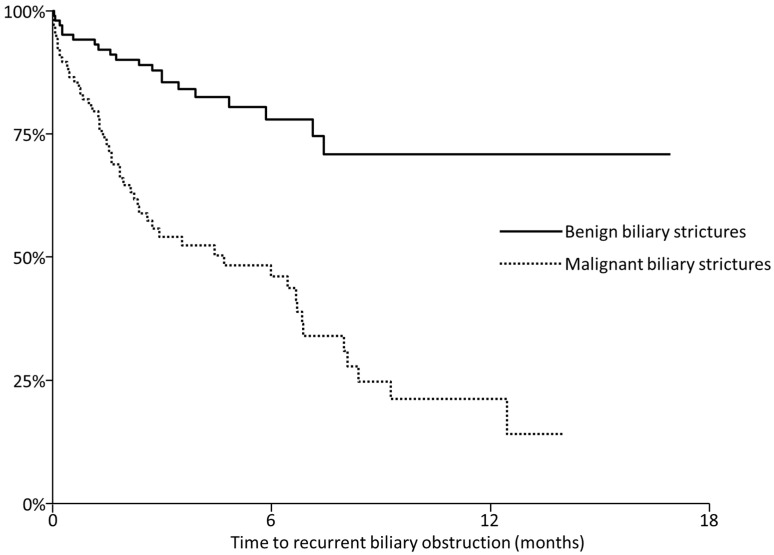
Kaplan-Meier curves of recurrent biliary obstruction in benign (solid line) and malignant (dotted line) biliary strictures.

**Figure 3 jcm-10-02936-f003:**
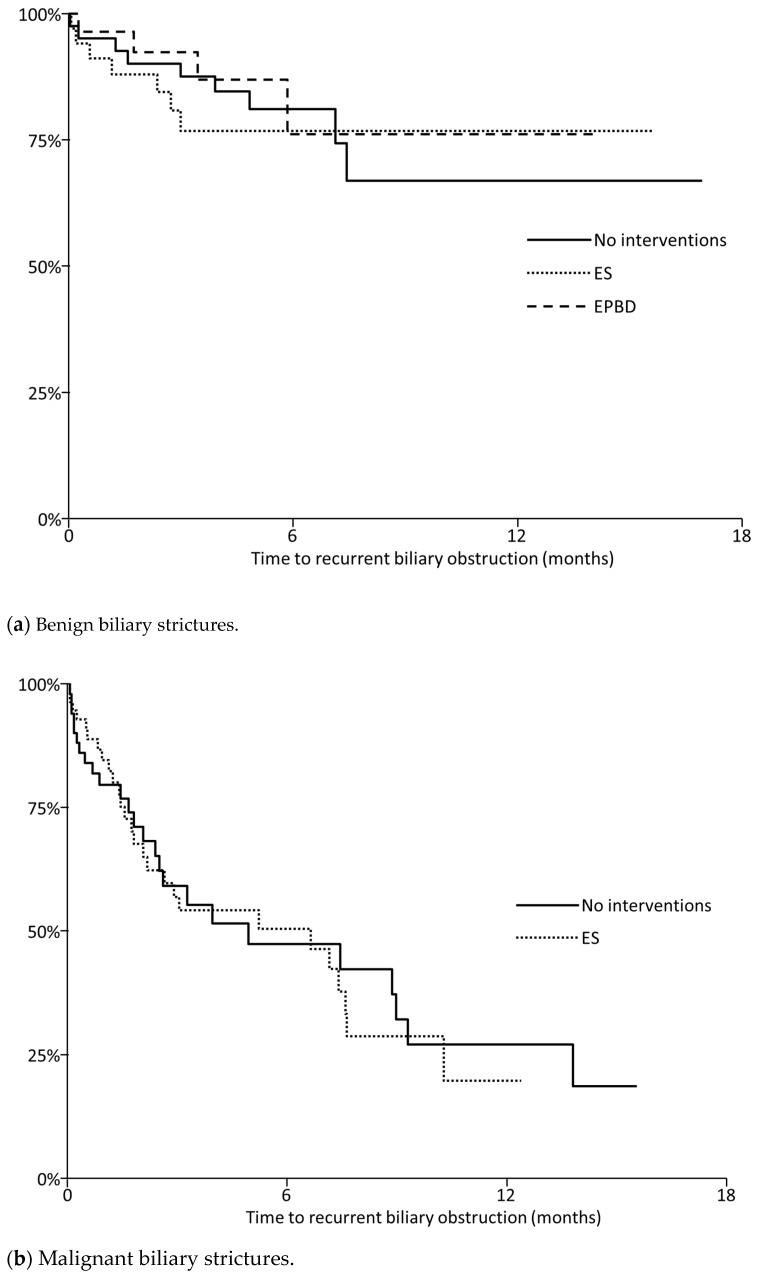
Kaplan-Meier curves of recurrent biliary obstruction according to the ampullary interventions in benign biliary strictures (**a**) and malignant biliary strictures (**b**). Solid line: No interventions, dotted line: ES (endoscopic sphincterotomy), Dashed line: EPBD (endoscopic papillary balloon dilation).

**Table 1 jcm-10-02936-t001:** Patient characteristics.

		Benign (*n* = 51)		Malignant (*n* = 81)
Age		63 (54–71)		73 (65–79)
Sex, Male		30 (59%)		41 (51%)
ASA PS 1/2/3		30/20/1		44/37/0
Etiology	Liver transplantation	31 (61%)	Bile duct cancer	36 (44%)
	Post-surgical stricture	7 (14%)	Gallbladder cancer	16 (20%)
	Primary sclerosing cholangitis	5 (10%)	Hepatocellular carcinoma	16 (20%)
	Others	8 (16%)	Others	13 (16%)
Concomitant treatment for malignancy		-	Surgical resection	10 (12%)
		-	Chemotherapy/radiation/TACE	40/4/12
Stricture site	Intrahepatic	15 (29%)		4 (5%)
	Hilar	27 (53%)		64 (79%)
	Common bile duct	9 (18%)		13 (16%)
Ampullary intervention	ES/EPBD/None	17 (33%)/9 (18%)/25 (49%)		40 (49%)/0/41 (51%)
Prior biliary drainage		35 (59%)		34 (42%)
Planned strategy	Planned exchange	41 (80%)		6 (7%)
	On demand exchange	7 (14%)		44 (54%)
	Temporary placement	1 (2%)		19 (23%)
	Unknown	2 (4%)		2 (2%)
White blood cell count		5890 (4600–7500)		6000 (4450–7740)
ALT		28 (15–46)		87 (55–177)
ALP		411 (299–945)		863 (547–1601)
Total bilirubin		0.8 (0.6–1.2)		4.1 (1.6–9.0)
C-reactive protein		0.42 (0.09–1.49)		1.59 (047–3.88)

Numbers are shown either in *n* (%) or median (interquartile ranges). ALP; Alkaline phosphatase, ALT; alanine transaminase, ASA PS; American society of anesthesiologists physical status, EPBD; endoscopic papillary balloon dilation, ES; endoscopic sphincterotomy, TACE; transcatheter arterial chemoembolization.

**Table 2 jcm-10-02936-t002:** Procedure details and clinical outcomes per session.

		Benign (*n* = 103)	Malignant (*n* = 106)
Balloon dilation of the stricture		39 (38%)	8 (8%)
The number of stents	1/2/3-	41 (40%)/53 (51%)/9 (9%)	64 (60%)/34 (32%)/8 (8%)
Stent size	7-Fr/8.5-Fr/Unknown	48 (47%)/54 (52%)/1 (1%)	48 (45%)/57 (54%)/1 (1%)
RBO	All	20 (19%)	52 (49%)
	Stent occlusion	13 (13%)	49 (46%)
	Stent migration	7 (7%)	3 (3%)
Adverse events	All	8 (8%)	8 (8%)
	Cholangitis without RBO	4 (4%): 4 mild	1 (1%): 1 mild
	Cholecystitis	1 (1%): 1 severe	4 (4%): 4 moderate
	Pancreatitis	3 (3%): 3 mild	3 (3%): 3 moderate

Numbers are shown in *n* (%). RBO; recurrent biliary obstruction.

## Data Availability

Data sharing is not applicable to this article.

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
