# Peer review of "A Prospective Multicenter Study of “Inside Stents” for Biliary Stricture: Multicenter Evolving Inside Stent Registry (MEISteR)"

_jcm, 2021, doi:10.3390/jcm10132936_

Round 1
Reviewer 1 Report
I enjoyed reading this manuscript entitled “A prospective multicenter study of "inside stents" for biliary stricture: Multicenter Evolving Inside Stent Registry (MEISteR)”.
This paper is interesting and evaluated the safety and effectiveness of the inside stents in patients with both malignant and benign biliary strictures as a prospective multicenter cohort study. During the follow-up period of 8.4 months, RBO was observed in 19% of benign strictures. The RBO rate was 49% in malignant strictures, with the median time to RBO of 4.7 months. The adverse event rate was 8%. I believe that these results will provide useful information to Journal of Clinical Medicine readers. Overall this manuscript is well-written, however, there is a few comment. The following comments should be addressed.
In conclusion, minor revise should be recommended for publication in Journal of Clinical Medicine.
<Minor comments>
- Regarding the results in the abstract, wouldn't it be better to list the primary endpoint “RBO” first rather than the technical success rates?
- In cases where no ES was performed at the time of insertion of the inside stent, was ES added at the time of removal of the stent?
- You described “The stricture resolution rate was 16% among 193 benign biliary strictures.” “The median duration 171 from the initial stent placement was 6.4 (IQR, 4.8-7.4) months.” “The median duration 171 from the initial stent placement was 6.4 (IQR, 4.8-7.4) months.” However, were the 8 patients whose stenosis improved after a single inside stent insertion? Did they improve after multiple stent replacements?
Author Response
We would like to thank the editor and reviewers for their thoughtful comments. We have revised our paper, and we believe our paper has improved thanks to the editor and reviewers.
Reviewer 1.
1. Regarding the results in the abstract, wouldn't it be better to list the primary endpoint “RBO” first rather than the technical success rates?
Response: As the reviewer recommended, we changed the results in Abstract and put the primary endpoint of RBO first.
We changed the sentences from “Technical success rates of stent placement and removal were both 100%. During the follow-up period of 8.4 months, RBO was observed in 19% of benign strictures. The RBO rate was 49% in malignant strictures, with the median time to RBO of 4.7 months.” to “During the follow-up period of 8.4 months, RBO was observed in 19% of benign strictures. The RBO rate was 49% in malignant strictures, with the median time to RBO of 4.7 months. Technical success rates of stent placement and removal were both 100%.”
2. In cases where no ES was performed at the time of insertion of the inside stent, was ES added at the time of removal of the stent?
Response: We did not perform ES to remove the inside stent in cases without prior ampullary interventions. We clarified this in Methods as follows.
“Additional ampullary interventions were unnecessary only for stent removal in cases without prior ampullary interventions.”
3. You described “The stricture resolution rate was 16% among 193 benign biliary strictures.” “The median duration 171 from the initial stent placement was 6.4 (IQR, 4.8-7.4) months.” “The median duration 171 from the initial stent placement was 6.4 (IQR, 4.8-7.4) months.” However, were the 8 patients whose stenosis improved after a single inside stent insertion? Did they improve after multiple stent replacements?
Response: Among 8 patients with stricture resolution, stricture resolution was obtained by a single session of inside stent placement only in 2 patients. In the remaining 6 patients, multiple ERCP sessions were necessary for stricture resolution. We added the following sentence in Results.
“Stricture resolution was obtained by a single session of inside stent placement in 2 patients, and the remaining 6 patients needed multiple sessions of inside stent placement for stricture resolution.”
Reviewer 2 Report
To the authors:
This is a well-organized manuscript regarding a prospective study of 132 patients who underwent inside stent for benign and malignant biliary strictures. This study investigated the safety and efficacy of inside stent. The primary endpoint was recurrent biliary obstruction (RBO), and secondary endpoints were the technical success rate of stent placement and removal, adverse events, and stricture resolution. RBO was observed in 20 (19%) of 103 endoscopic sessions in patients with benign strictures and 52 (49%) of 106 sessions in patients with malignant strictures. The technical success rate of stent placement and removal was 100%, while the adverse event rates were 8% in both patients with benign and malignant diseases. In addition, ampullary interventions such as endoscopic sphincterotomy or balloon dilation were performed in approximately a half of the patients; however, the interventions did not affect the time to RBO for both benign and malignant strictures.
Although, many retrospective studies have already demonstrated the superiority of inside stent over EBS in terms of stent patency, few prospective studies have evaluated the role of inside-stent. Thus, the topic of this study is interesting and potentially important. In addition, this study demonstrated that ampullary interventions such as sphincterotomy and balloon dilation did not affect the time to RBO, and also showed the technical difficulty of stent placement and stent removal in detail. This is a fascinating topic.
I think this paper should be accepted for publication in this journal when all my concerns are appropriately addressed in the revision.
(1) It is difficult to understand the results. Table 1 indicates patient demographics, and the population of each group is the number of patients. However, the results of RBO and adverse events, that is primary and secondary endpoints, are presented in Table 2, in which each population is the number of endoscopic sessions. The authors should describe the differences in the Methods section.
(2) Continuing from the previous comment, did RBO occur in 7 (14%) of 51 patients and 20 (19%) of 103 endoscopic sessions in the benign group? Is my understanding correct?
(3) Biliary stents are initially placed by local physicians. In resectable patients, such stents are subsequently replaced by other drainage methods in tertiary centers because of the need for further biliary evaluations. This study did not include initial biliary drainage methods or details of whether other biliary drainage procedures were used. It seemed that all the included patients underwent inside stent only from the initial drainage to the end of the study period.
Did the authors bring up inside stent only among several biliary drainage methods, that is, a plastic or metal stent across the papilla or ENBD?
(4) When RBO occurred in a patient with an inside stent, did the authors replace the inside stent with a new one or implement other drainage method, that is, ENBD, in principle?
(5) This study demonstrated that ampullary interventions such as sphincterotomy and balloon dilation did not affect time to RBO and also showed the details of technical difficulty at stent placement and stent removal. That is fascinating content.
Author Response
We would like to thank the editor and reviewers for their thoughtful comments. We have revised our paper, and we believe our paper has improved thanks to the editor and reviewers.
Reviewer 2.
1. It is difficult to understand the results. Table 1 indicates patient demographics, and the population of each group is the number of patients. However, the results of RBO and adverse events, that is primary and secondary endpoints, are presented in Table 2, in which each population is the number of endoscopic sessions. The authors should describe the differences in the Methods section.
Response: As the reviewer pointed out, Table 1 was per patient demographics, and Table 2 was per session analysis. Some patients received multiple sessions of inside stent placement due to the nature of the stricture etiology. We clarified this in Clinical outcomes in Methods as follows.
“Some patients received multiple sessions of inside stent placement, and all clinical outcomes were evaluated as per session analyses.”
2. Continuing from the previous comment, did RBO occur in 7 (14%) of 51 patients and 20 (19%) of 103 endoscopic sessions in the benign group? Is my understanding correct?
Response: RBO occurred in 20 sessions in 14 patients. We speculate 7 of 51 patients that the reviewer mentioned in the comment came from the number of patients of “on-demand exchange” in Table 1. The treatment strategy in Table 1 was not the result of inside stents. Rather, they were planned strategies. Thus, patients in planned stent exchange can develop RBO before planned procedures, while patients in on-demand stent exchange may not develop RBO. To clarify the number of patients with RBO in each group, we changed the sentence from “RBO was observed in 19% of benign strictures and 49% of malignant strictures.” to “RBO developed in 20 sessions (19%) in 14 patients with benign strictures and in 52 sessions (49%) in 37 patients with malignant strictures.”
In addition, we changed “strategy” to “planned strategy” in Table 1.
3. Biliary stents are initially placed by local physicians. In resectable patients, such stents are subsequently replaced by other drainage methods in tertiary centers because of the need for further biliary evaluations. This study did not include initial biliary drainage methods or details of whether other biliary drainage procedures were used. It seemed that all the included patients underwent inside stent only from the initial drainage to the end of the study period.
Did the authors bring up inside stent only among several biliary drainage methods, that is, a plastic or metal stent across the papilla or ENBD?
Response: We agree with the reviewer that many patients with biliary strictures have already been stented by primary physicians. Thus, we did not define the presence or absence of biliary drainage prior to inside stent placement as the inclusion criteria. As a result, about half of the patients had prior biliary drainage (Table 1). We added the following sentence to clarify this.
“Prior biliary drainage was present in 35 patients (59%) with benign strictures and 34 patients (42%) with malignant strictures. The prior biliary drainage was inside stents in 31, ENBD tube in 30 patients, conventional plastic stents across the papilla in 7 patients, and percutaneous transhepatic biliary drainage in 1 patient.”
4. When RBO occurred in a patient with an inside stent, did the authors replace the inside stent with a new one or implement other drainage method, that is, ENBD, in principle?
Response: In cases with cholangitis at RBO, the inside stent was replaced by ENBD first. After the resolution of cholangitis, ENBD was replaced by a new inside stent. We added the following sentence in “Placement and removal of inside stents” of Methods.
“In cases with cholangitis at the time of RBO, the inside stent was replaced by endoscopic nasobiliary drainage (ENBD) tube first if necessary, and after resolution of cholangitis, ENBD was replaced by a new inside stent or the other type of stents at the discretion of each physician.”
5. This study demonstrated that ampullary interventions such as sphincterotomy and balloon dilation did not affect time to RBO and also showed the details of technical difficulty at stent placement and stent removal. That is fascinating content.
Response: Thank you for your comments. We do agree that no effects on TRBO by the ampullary interventions are the key results of our study.